# The Role of Mitochondrial Homeostasis in Mesenchymal Stem Cell Therapy—Potential Implications in the Treatment of Osteogenesis Imperfecta

**DOI:** 10.3390/ph17101297

**Published:** 2024-09-29

**Authors:** Qingling Guo, Qiming Zhai, Ping Ji

**Affiliations:** 1College of Stomatology, Chongqing Medical University, Chongqing 401147, China; gqling@stu.cqmu.edu.cn; 2Chongqing Key Laboratory of Oral Diseases, Chongqing 401147, China

**Keywords:** osteogenesis imperfecta, mesenchymal stem cells, mitochondrial homeostasis, mitochondrial metabolism, antioxidants, mitochondrial quality control

## Abstract

Osteogenesis imperfecta (OI) is a hereditary disorder characterized by bones that are fragile and prone to breaking. The efficacy of existing therapies for OI is limited, and they are associated with potentially harmful side effects. OI is primarily due to a mutation of collagen type I and hence impairs bone regeneration. Mesenchymal stem cell (MSC) therapy is an attractive strategy to take advantage of the potential benefits of these multipotent stem cells to address the underlying molecular defects of OI by differentiating osteoblasts, paracrine effects, or immunomodulation. The maintenance of mitochondrial homeostasis is an essential component for improving the curative efficacy of MSCs in OI by affecting the differentiation, signaling, and immunomodulatory functions of MSCs. In this review, we highlight the MSC-based therapy pathway in OI and introduce the MSC regulation mechanism by mitochondrial homeostasis. Strategies aiming to modulate the metabolism and reduce the oxidative stress, as well as innovative strategies based on the use of compounds (resveratrol, NAD+, α-KG), antioxidants, and nanomaterials, are analyzed. These findings may enable the development of new strategies for the treatment of OI, ultimately resulting in improved patient outcomes.

## 1. Introduction

OI, also referred to as brittle bone disease, is a collection of hereditary conditions that predominantly impact bone integrity, leading to bones that are susceptible to breaking even from small injuries [1]. OI is caused by mutations in genes responsible for the production of type I collagen, a crucial protein that contributes to the strength and structure of bones as well as skin, ligaments, and tendons [2,3]. Most cases of OI are caused by mutations in the *COL1A1* and *COL1A2* genes, which are responsible for encoding the pro-α1 and pro-α2 chains of type I collagen, respectively [2,3]. These mutations can either decrease collagen production or result in the creation of aberrant collagen, ultimately leading to the weakening and brittleness of bones [4]. OI can be inherited from one or both parents, or it can arise from a spontaneous mutation in the affected individual [5]. Bisphosphonates are commonly used in the treatment of OI due to their ability to inhibit bone resorption by osteoclasts, thereby increasing bone density and reducing the risk of fractures [6]. However, the long-term use of bisphosphonates may result in the accumulation of microdamage, atypical femoral fracture, and mandibular osteonecrosis [7]. In addition to bisphosphonates, denosumab is an anti-resorptive drug that targets and inhibits RANKL, effectively reducing bone resorption and increasing bone density [8]. Teriparatide is an anabolic bone metabolic agent that stimulates bone formation by promoting osteogenesis, which is a different mechanism of action [9]. Another class of anabolic bone metabolic agents for the treatment of OI includes inhibitors of sclerostin or TGF-β. Sclerostin inhibitors, such as romosozumab, function by blocking the activity of sclerostin, leading to increased bone formation and strength [10]. Similarly, targeting TGF-β, a growth factor involved in bone remodeling, has been demonstrated to ameliorate high-turnover bone disease and enhance bone quality in OI models [11]. However, current treatments like bisphosphonates only partially correct the bone phenotype, and they have some disadvantages such as cytotoxic side effects, a lack of effect in certain patients, or relatively poor efficacy [11,12]. 

Stem cell therapy has the potential to enhance both the quality and quantity of bone in individuals with OI. The treatment of OI with MSCs offers a multifaceted approach through differentiation into bone cells, paracrine effects on bone tissue, and immunomodulatory actions [13,14,15]. Some studies suggest that extracellular vesicles (EVs) secreted by MSCs may be a crucial pathway for stem cell therapy in treating OI [16]. Although it has been confirmed that the transplantation of stem cells can improve the clinical symptoms of OI patients, the osteoblastic engraftment of donor cells was quite low [17,18]. We hope to find new strategies to enhance the function of mesenchymal stem cells, allowing a small number of implanted stem cells to exert a more significant effect, thereby achieving more notable and sustained symptom improvement in OI patients.

Mitochondria play an indispensable role in the mechanisms by which MSCs mediate the treatment of OI. During osteogenic differentiation, MSCs occur in metabolic reprogramming from glycolysis to oxidative phosphorylation (OXPHOS) [19]. This transition is accompanied by an increase in the production of new mitochondria, the replication of mitochondrial DNA (mtDNA), and the activation of respiratory enzymes. The mitochondria meet the increased energy demands in this process [19,20]. In addition, mitochondria control the generation of reactive oxygen species (ROS), which greatly influences the differentiation process of MSCs. Therefore, the maintenance of the function of MSCs is closely related to mitochondrial homeostasis [19,20]. 

In addition, the process of treating OI with MSCs through paracrine effects is partially mediated by mitochondrial dynamics. Mitochondria are involved in the secretion of various bioactive factors, such as IGF-1, VEGF, and BMP-2, which can stimulate bone growth and remodeling [21]. OI treatment can benefit from MSC-mediated immune responses that facilitate bone regeneration and repair, and mitochondria can play a role in these processes [15,22]. Mitochondrial transfer from MSCs to CD4+ T cells has been shown to inhibit their proliferation and reduce the secretion of pro-inflammatory cytokines, such as IFN-γ, thereby helping to create an immunosuppressive environment [23]. Based on these promising potential pathways, targeting mitochondrial pathways in MSCs represents a promising strategy for enhancing bone repair in OI [24,25]. 

We review the relevant research into MSC therapy for OI, focusing on the mechanism and application strategies of improving the function of MSCs, promoting bone regeneration and treating OI through the regulation of mitochondrial homeostasis in MSCs. 

## 2. MSC Therapy Mechanisms in OI

### 2.1. MSC Differentiation into Osteoblasts

MSCs are multipotent stromal cells capable of differentiating into various cell types, including osteoblasts, which are crucial for bone formation and remodeling. Osteoblasts synthesize and secrete the bone matrix and play a vital role in bone mineralization. This differentiation capacity makes MSCs particularly valuable in treating bone-related disorders like OI [26]. Sinder et al. demonstrated that bone marrow stromal cells (BMSCs) from donors can robustly engraft and differentiate into osteoblasts, generating healthy collagen in individuals with OI [27]. After three months of local transplantation, BMSCs showed a 12% increase in cortical thickness, a 14% rise in the polar moment of inertia, a 30% increase in bone strength, and a 30% increase in stiffness [27]. A recent research article demonstrated that the transplantation of hematopoietic stem cells (HSCs) in mice with OI resulted in the replacement of mutant osteoblasts with healthy osteoblasts that originated from HSCs [28]. This study shows significant improvements in bone morphometrics, mechanics, and turnover parameters [28]. Using a mouse model of OI, the transplantation of human fetal blood MSCs (hfMSCs) into the uterus led to an upregulation of osteogenic genes. These genes include osteocalcin, osteoprotegerin (OPG), Osterix (OSX), and bone morphogenetic protein 2 (BMP-2) [29]. These studies all demonstrate the promising prospects of using MSCs to treat OI, highlighting the importance of promoting MSCs’ osteogenic differentiation. In some clinical studies, the osteoblastic engraftment rate of donor cells after stem cell infusion was only 1–2%, yet patients with OI still experienced improvements in clinical symptoms following stem cell transplantation [17,18]. If we cannot increase the implantation rate of stem cells, can we enhance the potency of the stem cells to enable a small number of implanted MSCs to have a more significant effect?

MSCs achieve osteogenic differentiation through many signaling mechanisms, including bone morphogenetic proteins, as well as the Wnt and Notch pathways, which interact with each other and with transcription factors to regulate MSCs’ differentiation [30]. Transcription factors like Runx2 and OSX occupy a very important position in starting the process of osteogenic differentiation. BMP-2 stimulates the expression of Runx2 and OSX, which are responsible for guiding MSCs to become osteogenic cells and develop into mature osteoblasts [31]. Besides their role in MSCs’ osteogenic differentiation, these transcription factors also have intimate connections with mitochondrial function. Runx2 transcription factor has been shown to negatively regulate SIRT6, a member of the sirtuin family of proteins that regulate gene expression and are known to enhance mitochondrial activity [32]. SIRT6 enhances mitochondrial oxygen consumption, while RUNX2 inhibits SIRT6 expression and reduces oxygen consumption [32]. All of those show a strong correlation in the osteogenic differentiation of MSCs and mitochondrial function, and maintaining mitochondrial homeostasis is a feasible strategy to guide mesenchymal stem cells to achieve our purpose.

### 2.2. Paracrine Effects in Bone Healing

MSC therapy can improve OI through osteogenic differentiation. However, the efficacy of MSC therapy, regardless of the mechanism and delivery approaches, must consider the homing, adhesion, survival, retention, immunomodulation, angiogenesis, engraftment, and integration of transplanted MSCs at the tissue repair site [15,33]. Otsuru et al. conducted an early study in which they discovered that MSCs release a soluble mediator that indirectly promotes bone formation [34]. In the TERCELOI clinical trial, proteomics and transcriptomics analyses were employed to study two children with severe osteogenesis imperfecta who received multiple injections of MSCs isolated from HLA-matched siblings. They found that the expression of proteins controlled by the TGF-β and BMP signaling pathways increased to varying degrees, and there was also the upregulation of signaling molecules related to osteoprogenitor cells, collagen binding, and the extracellular matrix [26,35]. When MSCs are cultured in an osteogenic medium, they secrete distinct substances at certain times that enhance alkaline phosphatase activity (ALP) in external MSCs and facilitate their movement [36]. Additionally, factors released by bone marrow-derived mesenchymal stem cells (BM-MSCs) through paracrine secretion recruit macrophages and endothelial cells to the injured location [37]. During this process, wounds heal faster, as the cells migrate and multiply, and the presence of progenitor cells and immune cells increases [37]. Several studies have shown that MSCs release various types of EVs, including microvesicles (MVs) and exosomes that can serve as paracrine mediators between MSCs and target cells [38,39,40].

The release of EVs, MVs, and exosomes in MSCs is a complex process that mitochondria play an important role in this process. The release of EVs from MSCs is regulated by mitochondrial stress and dysfunction. Treating MSCs with mitochondrial inhibitors, such as rotenone and antimycin A, which impair mitochondrial function, results in an obvious change in the size distribution of the released EVs [41,42]. This suggests that mitochondrial function influences EV biogenesis and release. Similarly, MSCs cultured under physiological oxygen conditions release fewer EVs compared to cells with mitochondrial dysfunction induced by high oxygen or other stressors, which implies mitochondrial homeostasis regulates EV secretion [43,44]. Therefore, mitochondrial homeostasis is the foundation for the normal function of paracrine secretion in MSCs [45].

### 2.3. Immunomodulation by MSCs

The occurrence of OI may be related to inflammation. MSCs have immunomodulatory properties, which can regulate immune response, reduce the level of inflammation, and establish a conducive setting for bone healing, thus avoiding the exacerbation of bone fragility due to inflammation and damage to delay the healing process [13,46]. Zhang et al. found that MSCs can alter the cytokine profile in the body, elevating the levels of anti-inflammatory cytokines such as IL-10 and decreasing pro-inflammatory cytokines like IL-6 and TNF-α. Creating a favorable environment for bone regeneration can result from this shift [47]. MSCs inhibit T cell proliferation through direct contact and secrete specific cytokines like TGF-β, which promote the formation of regulatory T cells, aiding in a regulated inflammatory response crucial for bone repair [13]. Moreover, MSCs can influence the immune system by transferring their mitochondria to different immune cells, such as T cells. The transfer of MSC mitochondria to Th1 cells has been shown to reduce the expression of T-bet, which is the key transcription factor for Th1 cell development, leading to the suppression of IFN-γ production [23,48]. The transfer of MSC-derived mitochondria to Th17 cells can impair their IL-17 production and promote the generation of a significant suppressed regulatory T cell (Treg) population [21]. MSCs regulate immune cell function through metabolic reprogramming via mitochondrial transfer. This process enhances mitochondrial respiration in immune cells, leading to immunosuppressive effects, highlighting the importance of mitochondria in MSC-mediated immune regulation [49]. 

The three mechanisms of differentiation into osteoblasts, paracrine signaling, and immunomodulatory functions by which mesenchymal stem cells treat OI are briefly outlined in Figure 1.

## 3. Mechanisms of Mitochondrial Homeostasis Regulate the Function of MSCs

Research indicates that introducing human fetal blood MSCs (hfMSCs) into animal models of OI resulted in enhanced bone characteristics and reduced fracture occurrence [29,51]. In previous research, Horwitz et al. documented the clinical outcomes of three children with severe OI after receiving allogeneic mesenchymal bone marrow transplantation. A histological assessment conducted 216 days post-surgery, in addition to a bone densitometry analysis, showed significant improvement in bone structure. All patients had increases in total body bone mineral and growth velocity and reduced frequencies of bone fracture [52]. In the six months after transplantation, three children showed increased growth acceleration and total body bone mineral content in comparison to two non-transplanted OI controls. With extended follow-up periods up to six months after transplantation, growth rates slowed or plateaued, but the amount of bone mineral continued to rise [17]. Horwitz et al. treated six children with severe osteogenesis imperfecta using gene-marked, donor marrow-derived mesenchymal cells to show that mesenchymal cell treatment is feasible. In the first six months following infusion, engraftment was observed in five out of six patients, including the marrow stroma, skin, and bone. Additionally, there was an acceleration of growth velocity [18]. MSC therapy provides a feasible method to reduce the fracture rate in individuals with OI, which can be attributed to the donor cells differentiating into osteoblasts and generating normal bone matrix proteins, such as collagen type I [29,52]. However, Horwitz et al. also pointed out that although the symptoms of OI patients were improved after stem cell transplantation, the implantation rate of stem cells was low [17,18]. The question of how to increase the osteogenic differentiation of successfully implanted stem cells is particularly important, improving the function of MSCs by maintaining mitochondrial homeostasis may achieve this goal.

Mitochondrial homeostasis is implicated in numerous intricate cellular processes, including autophagy, the regulation of MSCs’ differentiation, and immunological responses [53,54]. These processes highlight the pivotal role of mitochondria in physiological regulation, cellular health, and disease. Mitochondrial homeostasis constitutes a prerequisite for the normal functioning of mitochondrial activities [55]. Mitochondrial homeostasis refers to the balance and stability of mitochondrial function, structure, and quantity within cells, ensuring the optimal performance of mitochondria and adaptation to constantly changing cellular demands and environmental conditions [56,57]. Mitochondrial homeostasis includes the complex processes of mitochondrial biogenesis, mitophagy, dynamics, proteostasis, signaling, and metabolic pathways that preserve the integrity and functionality of mitochondria [58]. Mitochondrial dysfunction can lead to MSC function decreases, especially osteogenesis differentiation [19,59]. During the osteogenic differentiation process, mitochondrial biosynthesis, oxygen consumption, and ATP production all increase, indicating the significant role of mitochondria in this process [60]. Mitochondrial dysfunction can result in the overproduction of ROS, leading to oxidative stress and negative impacts on the cellular environment, further affecting the osteogenic differentiation process of MSCs [61]. Maintaining mitochondrial redox balance is crucial for sustaining MSCs’ osteogenic differentiation, which is the basis of MSC therapy of OI [62] (Figure 2).

### 3.1. Mitochondrial Metabolism in MSCs

Mitochondrial metabolism is indispensable in meeting the diverse metabolic demands of MSCs. During osteoblast formation, there is a great rise in the rate of oxygen consumption and the levels of ATP within the cell, indicating a close association between mitochondrial energy metabolism and MSCs’ differentiation [21]. When stem cells age, damaged mitochondria exhibit abnormal OXPHOS, leading to excessive ROS production. The shift from OXPHOS to glycolysis decreases ATP generation, triggering AMPK activation, which leads to cell cycle arrest and promotes aging [64]. Regulating ROS levels can effectively maintain the functionality of MSCs and decline functional attenuation [65]. Compounds such as 17β-Estradiol have been found to decrease oxidative stress damage, enhance antioxidant enzyme activity, and protect MSCs from the negative effects of ROS induced by mitochondrial production under high-glucose conditions [66]. Furthermore, the PPAR-γ agonist pioglitazone has demonstrated the ability to shield MSCs from damage caused by oxidative stress through regulating ROS levels, indicating its potential in promoting bone regeneration and reducing the risk of osteoporosis [67]. The metabolites produced by the mitochondrial OXPHOS pathway are pivotal in regulating the activity, quiescence, self-renewal, and differentiation of MSCs by influencing ROS levels [21].

MSCs undergo osteogenesis differentiation through a complex interplay between metabolic byproducts and epigenetic regulation, which is driven by mitochondrial metabolism. Metabolites produced by mitochondria serve as essential substrates for various chromatin-modifying enzymes, therefore controlling gene expression in MSCs through enabling specific chromatin alterations [68]. These mitochondrial metabolites are instrumental in regulating gene expression by promoting the acetylation of β-catenin, a process that not only induces its nuclear translocation but also enhances downstream transcriptional activity master regulators such as Runx2 and OSX [69]. This regulatory mechanism plays an important role in directing the differentiation of MSCs towards osteoblasts rather than other cell lineages.

Moreover, mitochondrial metabolites like S-adenosylmethionine (SAM) and α-ketoglutarate (α-KG), which are essential for the OXPHOS process, influence the epigenetic landscape through N6-methyladenine (N6-mA) DNA modification [70]. This modulation of the epigenetic landscape is key to the fate determination of MSCs. The demethylase Alkbh1, regulated by N6-mA modification, has been identified as a significant participant in this regulatory mechanism [71]. The knockout of Alkbh1 has been shown to skew the differentiation potential of BMSCs toward an adipogenic lineage while concurrently inhibiting osteogenic differentiation [72]. This imbalance in differentiation potential underscores the importance of mitochondrial metabolism and epigenetic modifications to the compromised bone quality and increased fracture risk characteristic of OI. 

This intricate network of mitochondrial metabolism, epigenetic modifications, and gene expression regulation underscores the significance of mitochondrial pathways in the osteogenic differentiation of MSCs and their potential as therapeutic targets. By influencing key transcriptional regulators and modifying the epigenetic landscape, mitochondrial metabolites directly or indirectly impact the cellular fate decisions of MSCs. Targeting these mitochondrial pathways could offer new strategies for improving bone quality and reducing the fracture risk in OI, emphasizing the necessity for additional investigation into mitochondrial function and its therapeutic potential in OI [68,73,74].

### 3.2. Mechanism of Mitochondrial Anti-Oxidative Stress

The presence of ROS is pivotal in bone regeneration and reconstruction, but their levels must be finely regulated to ensure optimal therapeutic outcomes. Appropriate levels of ROS are necessary for various signaling pathways involved in bone formation; excessive ROS can impair stem cell function and hinder bone healing processes [75,76]. As MSCs differentiate into osteoblasts, there is a decrease in intracellular ROS levels, accompanied by an upregulation of antioxidant enzymes including superoxide dismutase (SOD), catalase (CAT), and glutathione peroxidase (GPx) [77]. This upregulation of antioxidant enzymes helps counteract the adverse impacts of excessive ROS and promotes osteoblast differentiation and bone reconstruction [78] (Figure 3).

Antioxidants and natural enzymes contribute to the control of oxidative signals and scavenging of excessive ROS in MSCs, thus modulating their behaviors and fate decisions [80,81]. A myriad of papers have indicated that the osteogenic capacity of MSCs could be improved by either providing antioxidant supplementation or overexpressing the antioxidant enzymes [82,83,84]. Nevertheless, it should be pointed out that total ROS depletion is not desirable as regulated ROS levels are needed for cellular signaling and the activation of pathways associated with osteoblast differentiation and bone matrix production [85]. Thus, delicate control over ROS levels would be essential to promote bone regeneration and reconstruction and, on the other hand, to avoid increasing oxidative stress to harmful levels.

These results show that targeting ROS modulation and stimulating antioxidant defense mechanisms in MSCs could be an effective strategy in the MSC treatment of OI [76]. Targeting oxidative stress and antioxidants, therefore, has the potential to fine-tune the redox balance and facilitate osteoblast differentiation to improve bone quality and healing in OI [76].

### 3.3. Mitochondrial Quality Control in MSCs

Mitochondrial quality control (MQC) systems are essential for maintaining cellular homeostasis and ensuring the normal function of MSCs during bone-forming differentiation. These systems encompass various mechanisms, including protein balance, generation, movement, and mitophagy [86]. The accumulation of mitochondrial defects may impair cellular function and differentiation, and MQC is indispensable for preventing the elimination of damaged or dysfunctional mitochondria [87]. For the MSC treatment of OI, MQC mechanisms are expected to be potential therapeutic targets for enhancing the osteogenic potential of MSCs [45,88].

The NAD+-dependent deacetylase family, sirtuin, which is involved in mitochondrial quality control, has been regarded as a novel therapeutic target for the enhancement of bone quality, highlighting its role in the osteogenesis of MSCs [89]. SIRT1 has been proven to enhance the process of MSCs transforming into bone cells by controlling the creation and operation of mitochondria [88,90]. Furthermore, resveratrol has been found to improve the osteogenic differentiation of aging-affected PO-MSCs (periosteum-derived MSCs) by upregulating the expression of mitochondrial inner membrane proteins (such as Mitofilin) [91]. Research has shown that resveratrol greatly improves extracellular matrix calcification and the expression of osteogenic-related genes, inhibits ROS production in BMSCs, and upregulates AMPK expression, suggesting its potential mechanism in promoting BMSC osteogenic differentiation [92,93].

The mTOR signaling pathway also regulates mitochondrial quality control by affecting mitochondrial biogenesis, dynamics, and autophagy [94]. Rapamycin, an inhibitor of mTOR signaling, upregulates the expression of genes involved in mitochondrial autophagy and mitochondrial fission, while downregulating genes associated with mitochondrial fusion (such as OPA1), thereby maintaining mitochondrial homeostasis [95]. During osteogenic differentiation, an appropriate concentration of rapamycin promotes BMSC osteogenic differentiation through autophagy activation [96].

Collagen membranes crosslinked with 3,4-dihydroxyphenylacetic acid (HPAA-HCM) have been demonstrated to enhance the osteogenic development of MSCs [97]. The self-mineralization ability of these membranes provides a bone-conductive scaffold, supporting the growth and differentiation of MSCs into osteoblasts [98]. HCM accelerates osteogenic differentiation by activating mitochondrial dynamics, which is associated with the expression of key proteins regulating mitochondrial morphology and function, highlighting its role as a core mechanism in osteogenic differentiation [94,97].

In conclusion, MQC mechanisms, such as the regulation of mitochondrial biogenesis, dynamics, and autophagy, are pivotal in the osteogenic differentiation of MSCs and offer potential therapeutic targets for treating OI [86,94]. Sirtuins, resveratrol, mTOR signaling, and HPAA-HCM could become promising approaches to modulate MQC and enhance the osteogenic potential of MSCs.

## 4. Strategies for Regulating the Function of MSCs through Mitochondrial Homeostasis

### 4.1. Regulating Mitochondrial Metabolism in MSCs

#### 4.1.1. Resveratrol

Resveratrol, a naturally occurring polyphenolic compound, has been identified as a potent modulator of mitochondrial function and metabolic homeostasis [89,91,94]. It exerts its effects by activating SIRT1 (NAD+-dependent deacetylase), which in turn promotes the deacetylation and activation of PGC-1α. PGC-1α is a key controller of mitochondrial biogenesis [89,92]. This activation cascade enhances mitochondrial function and may counteract the mitochondrial dysfunction associated with OI.

Resveratrol’s therapeutic potential not only promotes mitochondrial biogenesis but also has anti-inflammatory properties that are beneficial in treating inflammatory diseases [93,95]. Studies have demonstrated that resveratrol can promote osteogenic differentiation in various MSC populations, such as gingival MSCs and BMSCs, and can ameliorate age-related osteoporosis by restoring mitochondrial OXPHOS capacity through pathways involving Mitofilin and PGC-1α [96,97]. In osteogenic cultures, resveratrol treatment has been shown to enhance ALP, an early indicator of osteogenic differentiation, and to promote mineralization, a late-stage indicator of bone formation [75,76]. These findings indicate that resveratrol positively influences bone formation and may have therapeutic benefits which enhance bone quality and decrease fracture risk in individuals with OI. The direct supplementation of mitochondrial metabolites, such as resveratrol, offers a novel approach for therapeutic research, potentially leading to innovative treatments for metabolic and degenerative diseases such as OI [95]. By targeting mitochondrial pathways, resveratrol and similar compounds could provide a means to enhance the regenerative capacity of MSCs.

#### 4.1.2. NAD+

Nicotinamide adenine dinucleotide (NAD+), an essential coenzyme in OXPHOS, has been shown to significantly influence MSC differentiation [98]. During the osteogenic differentiation of BMSCs, a marked metabolic shift occurs, characterized by an upregulation of OXPHOS activity, a reduction in glycolysis, and an increase in intracellular NAD+ levels [99,100]. This shift from glycolysis to OXPHOS is necessary for meeting the increased energy demands of differentiation and bone formation. The inhibition of NAD+ synthesis disrupts this metabolic reprogramming, leading to mitochondrial dysfunction and impaired osteogenic differentiation [80]. Conversely, replenishing NAD+ levels can partially restore this impairment, underscoring the crucial importance of NAD+ in supporting mitochondrial function and osteogenesis [101]. Elevated NAD+ levels are crucial for maintaining important metabolic pathways, including glucose glycolysis, the tricarboxylic acid (TCA) cycle, and fatty acid β-oxidation, which are necessary for cellular metabolism and energy generation [102,103]. This highlights the indispensable role of NAD+-mediated mitochondrial OXPHOS in BMSC osteogenic differentiation and provides insights into the mechanisms of bone formation, suggesting potential therapeutic targets for enhancing bone repair and regeneration.

Furthermore, maintaining optimal NAD+ levels has emerged as a novel therapeutic target in regenerative medicine. Studies have demonstrated that the exogenous supplementation of NAD+ can effectively delay senescence in BMSCs induced by external factors, such as D-galactose, and increase intracellular NAD+ levels [81]. The silencing of SIRT1, an NAD+-dependent deacetylase, exacerbates D-galactose-induced senescence, highlighting the defensive impact of exogenous NAD+ on aging BMSCs [84,104]. This suggests that interventions aimed at enhancing NAD+ levels and SIRT1 activity could offer promising strategies for combating an age-related decline in BMSC function and promoting bone regeneration in conditions like OI [105,106].

#### 4.1.3. Alpha-Ketoglutarate (α-KG)

α-KG is a key intermediate in the TCA cycle that plays a crucial role in mitochondrial metabolism. Emerging evidence suggests that α-KG has therapeutic potential for bone disorders [107,108]. α-KG exerts its beneficial effects on MSCs by regulating the epigenetic landscape, specifically through modulating histone methylation patterns. It reduces the enrichment of repressive histone marks, such as H3K9me3 and H3K27me3, on the promoters of key osteogenic genes like BMP2, BMP4, and Nanog [87]. By alleviating this epigenetic repression, α-KG promotes the expression of these genes, thereby stimulating the proliferation, migration, and osteogenic differentiation of MSCs [87].

In vivo research has shown the therapeutic potential of α-KG supplementation in combating osteoporosis and age-related bone loss. The administration of α-KG significantly attenuates bone deterioration in aged mice, preserving bone mass and strength [87,109]. Moreover, α-KG accelerates bone regeneration in aged rodents, highlighting its capacity to improve the regenerative capacity of MSCs and promote bone repair [110]. Beyond its epigenetic effects, α-KG is essential in supporting bone matrix formation and collagen synthesis [111]. As a key intermediate in the TCA cycle, it serves as a significant source of amino acids required for collagen production in cells and organisms [112]. This metabolic function of α-KG helps maintain the integrity of bone tissue and the promote osteogenesis.

Furthermore, α-KG was demonstrated to exert pro-osteogenic effects in osteoblast cell lines through the activation of signaling pathways such as JNK and mTOR/S6K1/S6 [113]. These pathways are involved in regulating cell proliferation, differentiation, and protein synthesis, further supporting the significance of α-KG in promoting bone formation [114]. In conclusion, mitochondrial metabolism, particularly the TCA cycle intermediate α-KG, is crucial in the osteogenic differentiation of MSCs and has substantial therapeutic potential in the treatment of OI and other bone disorders. These relevant studies are summarized in Table 1. 

### 4.2. Mitochondrial Anti-Oxidative Stress Strategy through Antioxidants

#### 4.2.1. N-Acetylcysteine (NAC)

NAC and other antioxidants are essential in preserving cellular homeostasis and supporting the function of MSCs by scavenging ROS. Yamada et al. found that NAC can increase the expression of bone-related genes such as collagen I, osteopontin, osteocalcin, BMP-2, and Runx2/cbfa1, which implies NAC accelerated bone regeneration by activating the differentiation of osteogenic lineages [111]. In a study involving rat femoral defects, the pre-treatment of autologous BMSCs with NAC, followed by implantation using a collagen sponge, yielded markedly higher new bone growth in comparison to the control group [112]. Various studies have shown that NAC has indeed been shown to significantly reduce ROS levels and promote fracture healing [76,113,114]. NAC has been shown to promote bone regeneration through various mechanisms, including its antioxidant activity, its ability to enhance osteogenic differentiation, the activation of pro-osteogenic signaling pathways, improvements in scaffold properties, and the promotion of bone healing in vivo. These properties of NAC suggest its potential therapeutic application in the context of OI. 

#### 4.2.2. Vitamin C

Vitamin C, another potent antioxidant, is essential for osteoblast growth and differentiation, making it a key player in musculoskeletal healing. Clinical studies have demonstrated that patients undergoing open reduction and internal fixation surgery who receive vitamin C treatment show increased plasma levels of ALP and osteocalcin, which are markers of increased bone density and faster healing [115]. Roman et al. demonstrated that vitamin C plays a role in orchestrating osteogenic differentiation and function by modulating chromatin accessibility and priming transcriptional activity through epigenetic mechanisms [116]. These findings highlight the advantageous impacts of vitamin C supplementation on bone regeneration and repair.

#### 4.2.3. Alpha-Lipoic Acid (α-LA)

α-LA, a powerful mitochondrial antioxidant, is gaining recognition for its role in maintaining cellular redox balance and promoting bone tissue regeneration. α-LA can protect against bone loss in the rat mandible and promote bone formation by inhibiting oxidative stress [117]. Similarly, another study found that the supplementation of α-LA promotes the healing of femoral fractures in rats [118]. α-LA therapy enhances bone regeneration in bone loss patients by increasing new bone formation, bone volume, and bone mineral density in vivo. α-LA triggers signaling pathways that promote osteogenic differentiation and bone production, such as the PI3K/AKT pathway [119,120]. OI and osteoporosis are both conditions characterized by bone fragility; they have different underlying causes and clinical features, but they have some similarities in terms of treatment, which shows the potential promise of using α-LA to treat OI. The modulation of these signaling pathways could help counteract the impaired osteogenesis observed in OI. These relevant studies have been summarized in Table 2.

### 4.3. Mitochondrial Anti-Oxidative Stress Strategy through Biomaterials

#### 4.3.1. Graphene Oxide (GO)

GO is a modified form of graphene that has been functionalized with several oxygen-containing groups, showing properties of being hydrophilic and biocompatible. Graphene oxide, when modified with metal nanoparticles, has been demonstrated to improve the anti-inflammatory properties in MSCs for tissue engineering purposes [129]. This characteristic is especially significant in the setting of MSC therapy, where increased oxidative stress and inflammation contribute to the impaired osteogenic differentiation of MSCs and defective bone formation [80].

#### 4.3.2. Fullerene Alcohol/Alginate Hydrogels

Fullerene alcohol/alginate hydrogels are injectable cell delivery carriers with antioxidant properties. These hydrogels can effectively scavenge superoxide anions and hydroxyl radicals, protecting brown adipose-derived stem cells from oxidative stress and enhancing their survival in ROS-rich environments [129]. Similarly, polydopamine (PDA), a material with catechol functional groups, possesses antioxidant properties and has been used to fabricate nanostructures that protect MSCs from oxidative stress [130]. PDA and its derivatives have been utilized to enhance stem cell functionality, remove ROS, and alleviate oxidative stress [131]. By scavenging excessive ROS and mitigating oxidative stress, these nanostructures create a favorable microenvironment for MSC survival and differentiation. 

#### 4.3.3. Polydopamine (PDA)

PDA-coated substrates have been shown to significantly reduce oxidative stress and mitochondrial damage in replicative senescent MSCs. This is in part due to the scavenging of extracellular ROS, an aggravating factor of cellular senescence [132]. In addition, the PDA coating may also reduce the expression of senescence-associated genes, including p53 and p21, while enhancing the expression of stemness-associated genes such as OCT4 [133]. PDA nanoparticles, due to their distinct hydroquinone moiety, possess enhanced antioxidant properties, which allows them to scavenge free radicals effectively. This property is crucial for protecting MSCs from oxidative stress-induced damage [134,135]. 

#### 4.3.4. Cerium Oxide Nanoparticles (CeNPs)

CeNPs demonstrate catalytic behavior by scavenging ROS through redox cycles between Ce3+ and Ce4+ states via electron charge transfer, collecting, storing, or releasing oxygen on their surface [136,137]. This unique feature allows CeNPs to effectively regulate the concentration of ROS in cellular environments, providing a protective effect [138,139]. By utilizing a hydrothermal method to construct a microenvironment-responsive bio-functional metal–organic framework (bio-MOF) coating on titanium surfaces through coordination between pxylylenebisphosphonate (PXBP) and Ce/Sr ions, catalytic properties similar to CAT and SOD are achieved, decomposing ROS in MSCs and restoring their mitochondrial function [136]. This method has demonstrated encouraging outcomes in stimulating the osteogenic differentiation of MSCs and improving bone repair [140]. 

#### 4.3.5. Manganese Dioxide

The antioxidant mechanism of manganese dioxide (MnO2-x) nanoparticles is attributed to their enzyme-like activity, acting as catalysts to remove superoxide and peroxynitrite, protecting cells from oxidative stress [141]. Manganese-containing bioceramics, such as manganese-doped β-tricalcium phosphate (Mn-TCP), can inhibit osteoclast formation by clearing ROS, promoting bone regeneration in osteoporotic bone defects [142]. These findings suggest that MnO2-x nanoparticles could potentially be employed to modulate the osteogenic differentiation of MSCs and improve bone quality in OI by regulating ROS levels and mitochondrial function [132]. 

#### 4.3.6. Iron Oxide

Iron oxide (Fe_3_O_4_) nanoparticles can be prepared with natural antioxidant activity by incorporating plant or fruit extracts, enhancing their bioactive functions and providing a sustainable method for the environmentally friendly preparation of metal nanoparticles [143,144]. These green-synthesized Fe_3_O_4_ nanoparticles have shown promising results in scavenging ROS and promoting the osteogenic differentiation of MSCs [145,146]. By modulating mitochondrial metabolism and reducing oxidative stress, Fe_3_O_4_ nanoparticles could potentially be used to treat OI by enhancing the osteogenic potential of MSCs and improving bone formation [84].

Nanomaterials with enzyme-like capabilities offer distinct advantages over natural enzymes, such as enhanced stability, cost-effectiveness, and easy storage, making them well suited for biomedical uses [147,148,149]. These nanomaterials, also known as nanozymes, show promising prospects in the treatment of OI due to their potential in scavenging ROS and regulating the cellular microenvironment [132,150]. These relevant studies have been summarized in Table 3.

## 5. Conclusions

Exploring MSC therapy for OI by controlling mitochondrial homeostasis is a prospective way of curing this genetic disorder. Current research has demonstrated that MSC therapy can improve clinical symptoms in patients with OI. However, the implantation rate of stem cells remains relatively low. By summarizing previous studies, it has been suggested that maintaining mitochondrial homeostasis to enhance the function of MSCs may allow even a small number of implanted stem cells to have a more significant impact, potentially enhancing the efficacy of MSC therapy for OI patients. Mitochondrial metabolic modulation, anti-oxidative stress approaches, and the application of nanomaterials offer innovative ways of maintaining mitochondrial homeostasis, enhancing bone regeneration, and treating OI. The potential of compounds like resveratrol, nicotinamide adenine dinucleotide (NAD+), and α-KG in regulating mitochondrial function and osteogenic differentiation of MSCs is particularly noteworthy. Furthermore, the role of antioxidants and nanomaterials in protecting MSCs from oxidative stress and improving bone healing outcomes creates new possibilities for OI therapy. Mitochondrial quality control mechanisms, such as protein homeostasis, biogenesis, dynamics, and mitophagy, are indispensable in maintaining cellular homeostasis and ensuring the proper function of MSCs during osteogenic differentiation. Targeting mitochondrial pathways could offer new strategies for improving bone quality and reducing fracture risk in OI patients, emphasizing the need for further research into mitochondrial function and its therapeutic potential in OI. By focusing on the regulation of mitochondrial function, researchers and clinicians can potentially unlock new therapeutic strategies that not only address the symptoms of OI but also target its underlying molecular defects. As we advance our understanding of mitochondrial biology in the context of stem cell therapy, the prospects for developing effective treatments for OI and other bone-related disorders become increasingly promising.

## Figures and Tables

**Figure 1 pharmaceuticals-17-01297-f001:**
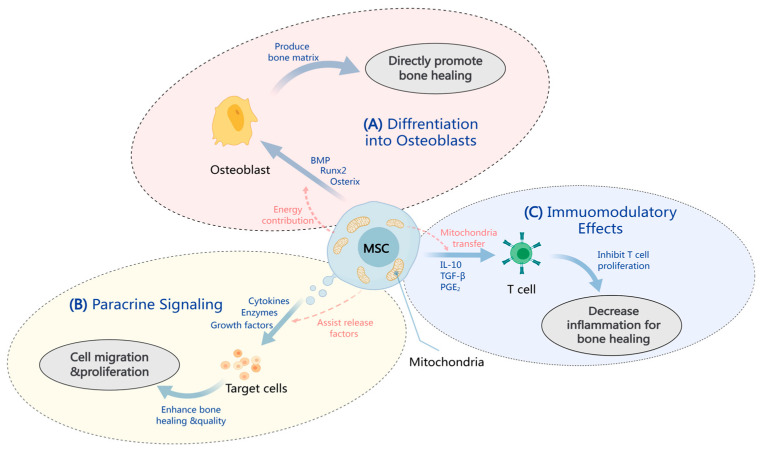
MSCs’ treatment of OI is multifaceted and involves several aspects of MSC biology, including their ability to differentiate into osteoblasts, their immunomodulatory effects, and their capacity to secrete bioactive molecules that can influence the bone healing process. (**A**) Differentiation into osteoblasts: BMP-2 stimulates the expression of Runx2 and OSX, and mitochondria provide energy for this process, promoting the differentiation of mesenchymal stem cells into bone [50]. (**B**) Paracrine signaling: MSCs release soluble mediators (cytokines, enzymes, and growth factors) that promote cell migration and proliferation to enhance bone healing and quality [14,33]. (**C**) Immunomodulatory effects: MSCs alter the cytokine profile in the body, elevating anti-inflammatory cytokines and decreasing pro-inflammatory cytokines [13].

**Figure 2 pharmaceuticals-17-01297-f002:**
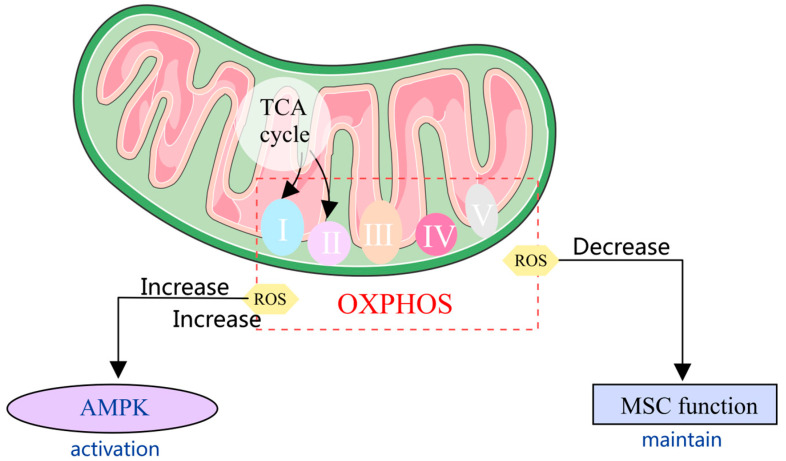
Mitochondrial metabolism is indispensable in meeting the diverse metabolic demands of MSCs. An increase in reactive oxygen species (ROS) generated by OXPHOS leads to the activation of AMP-activated protein kinase (AMPK). A decrease in ROS levels is associated with the maintenance of MSC function [63,64].

**Figure 3 pharmaceuticals-17-01297-f003:**
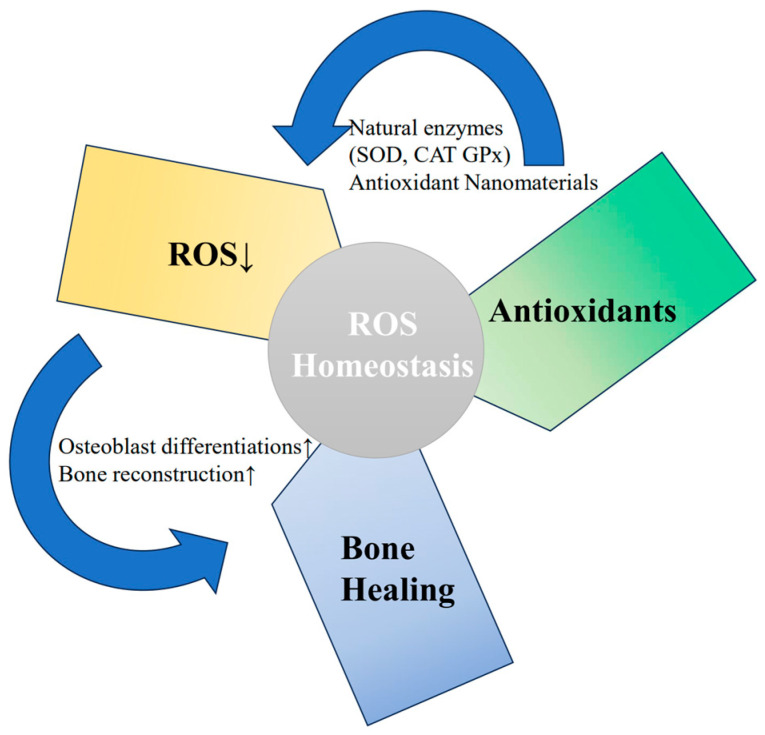
ROS are crucial for bone regeneration and reconstruction, but their levels must be regulated for optimal therapeutic outcomes. Excessive ROS can impair stem cell function and hinder healing. As MSCs differentiate into osteoblasts, ROS levels decrease and antioxidant enzymes upregulate, promoting osteoblast differentiation and reconstruction [75,76,79].

**Table 1 pharmaceuticals-17-01297-t001:** Strategies for regulating mitochondrial metabolism.

Metabolic Modulation	Summary	Key Mechanism	Biological Effect	Ref.
Resveratrol	Resveratrol can maintain the therapeutic potential of MSCs during long-term culture by acting through the SIRT1-SOX2 axis	SIRT1-SOX2 Axis	Improved bone regeneration	[74]
	Resveratrol promotes osteogenic differentiation and mitochondrial biogenesis in periosteum-derived MSCs	Mitochondrial Biogenesis	Enhanced osteogenesis	[75]
	Runx2 acetylation/deacetylation is a main mechanism during osteogenic differentiation in MSCs in vitro	Sirt-1/Runx2 Pathway	Promoted osteogenic differentiation	[76]
	Resveratrol enhances osteogenesis in human MSCs by upregulating the expression of the RUNX2 gene through the SIRT1/FOXO3A pathway	SIRT1/FOXO3A Axis	Enhanced osteogenesis	[77]
	Resveratrol enhances the proliferation and osteoblastic differentiation of human MSCs through ER-dependent ERK1/2 activation	ERK1/2 Activation	Increased proliferation and differentiation	[78]
	Resveratrol can attenuate osteoporosis by promoting the osteogenic differentiation of bone marrow MSCs through the SIRT1/PI3K/AKT pathway	SIRT1/PI3K/AKT Pathway	Attenuation of osteoporosis	[80]
NAD+	NAD+ levels affect osteoblastogenesis in cells from old mice, showing that reduced NAD+ impairs mineralization under osteogenic conditions	NAD+ Level Impact	Impaired mineralization	[81]
	NAD+ levels impair mitochondrial fusion, leading to mitochondrial dysfunction and reduced activity of OXPHOS, which subsequently blocks osteogenesis and diminishes bone fracture healing	Mitochondrial Dysfunction	Blocked osteogenesis and fracture repair	[82]
	Exogenous NAD+ can delay senescence in bone marrow-derived MSCs through the activation of the Sirt1 signaling pathway	Sirt1 Signaling Activation	Delayed senescence in MSCs	[83]
	miR-34a uses the NAD+-Sirt1 pathway to further mediate its role in MSC replicative senescence and natural senescence by targeting Nampt	NAD+-Sirt1 Pathway	Ameliorated MSC senescence	[84]
α-KG	α-KG promotes alveolar bone regeneration following jawbone injury by modulating macrophage polarization towards an M2 phenotype, which is conducive to healing and tissue repair	Modulation of Macrophage Polarization	Enhanced bone regeneration	[85]
	α-KG supplementation increases bone mass in aged mice and accelerates bone regeneration by decreasing histone methylations and upregulating BMP signaling and Nanog expression	Regulation of Histone Modifications	Accelerated bone regeneration	[86]
	α-KG influences stem cell fate and promotes osteogenic differentiation through mitochondrial nuclear signaling	Mitochondrial Signaling	Promoted osteogenic differentiation	[68]

**Table 2 pharmaceuticals-17-01297-t002:** Summary of antioxidant strategies of different antioxidants.

Antioxidants	Summary	Main Mechanism	Biological Effect	Ref.
NAC	NAC and ascorbic acid protect MSCs from oxidative stress-induced mitochondrial dysfunction by enhancing mitochondrial fusion and reducing fragmentation	Mitochondrial Protection	Enhanced mitochondrial function	[121]
	NAC inhibit ROS production and rescue MSCs from senescence by improving mitochondrial function and reducing oxidative stress	Mitochondrial Protection	Rescued MSCs from senescence	[122]
	NAC and mitochondria-targeted ubiquinone can reduce oxidative damage and improve the survival and differentiation of MSCs	Reduction in ROS	Improved survival and differentiation	[123]
	NAC and pyrrolidine dithiocarbamate reduce intracellular ROS and their effects on MSC chondrogenesis	Reduction in ROS	Reduced oxidative stress in chondrogenesis	[124]
α-LA	α-LA has potential effects on MSCs by protecting them from oxidative stress	Enhancement of Antioxidant Mechanisms	Protected MSCs from oxidative stress	[125]
	α-LA can protect mitochondria from oxidative stress by enhancing cellular antioxidant mechanisms	Enhancement of Antioxidant Mechanisms	Protected mitochondria from oxidative stress	[126]
Vitamin C	Vitamin C hydrogel scaffolds enhance cell survival and minimize ROS levels under H_2_O_2_-induced oxidative stress conditions	Free Radical Scavenging	Improved cell survival under oxidative stress	[127]
	Vitamin C can protect MSCs from oxidative stress-induced mitochondrial dysfunction	Mitochondrial Protection	Protected MSCs from oxidative stress-induced mitochondrial dysfunction	[128]

**Table 3 pharmaceuticals-17-01297-t003:** Strategy of scavenging ROS by new biomaterials.

Biological Material	Summary	Main Mechanism	Biological Effect	Ref.
GO	GO has potential to mitigate cadmium-induced cytotoxicity and oxidative stress	Mitigation of cadmium-induced cytotoxicity	Reduced oxidative stress	[151]
	GO exposure leads to significant decreases in mitochondrial membrane potential and ATP production	Mitochondrial dysfunction and ATP reduction	Decreased mitochondrial membrane potential and ATP generation	[152]
Fullerenol/Alginate Hydrogel	Fullerenol/alginate hydrogel can effectively scavenge superoxide anion and hydroxyl radicals, improving the survival of stem cells under oxidative stress	Antioxidant activity and cell delivery	Suppression of oxidative stress damage in MSCs	[129]
	Fullerene/alginate hydrogels in bone regeneration strategies bond to modulation of mitochondrial function and redox homeostasis	Organelle homeostasis and bone regeneration	Improved bone regeneration through organelle homeostasis of MSCs	[74]
PDA	PDA-coated substrate can reduce oxidative stress and mitochondrial damage in mesenchymal stem cells, enhancing their expansion and reducing senescence	Antioxidant properties and cellular protection	Reduction in oxidative stress and mitochondrial damage in MSCs	[133]
	PAD nanoparticles have enhanced antioxidant properties and cellular uptake, which could be beneficial for protecting MSCs from oxidative stress	Antioxidant effects and mitochondrial health	Enhancement of antioxidant properties and cellular uptake	[153]
CeNPs	CeNPs support the mitochondrial health of MSCs in regenerative contexts	Antioxidant and anti-inflammatory effects	Potential applications in wound healing and tissue regeneration	[154]
	CeNPs has potential in mitigating oxidative stress and protecting mitochondrial function in various cell types	Biocompatibility and cytotoxicity	Reduction in ROS levels and protection against oxidative stress	[155]
Fe_3_O_4_ nanoparticles	Iron oxide nanoparticles could augment intercellular mitochondrial transfer from MSCs	Oxidative stress and cytotoxicity	Enhanced intercellular mitochondrial transfer from MSCs to diseased cells	[156]
	Fe_3_O_4_ nanoparticles can be used for magnetic targeting and delivery of mesenchymal stem cells, improving their retention and therapeutic effects	Magnetic targeting and cell delivery	Improved cell retention and therapeutic effects in various disease models	[157]

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
