# Peer review of "The Role of Mitochondrial Homeostasis in Mesenchymal Stem Cell Therapy—Potential Implications in the Treatment of Osteogenesis Imperfecta"

_pharmaceuticals, 2024, doi:10.3390/ph17101297_

Round 1

Reviewer 1 Report

Comments and Suggestions for Authors

The Review “Advancing Osteogenesis Imperfecta Treatment: The Role of Mitochondrial Homeostasis in Mesenchymal Stem Cells Therapy” by Qingling G, Qiming Zhai and  Ping Ji highlight MSC-based therapy pathway in OI and introduce the 16 MSCs regulation mechanism by mitochondrial homeostasis.

The review is complete and well written, however it could be improved:

In the introduction, the authors should investigate the effect of bisphosphonates on OI and should better clarify the role of inhibitors of sclerostinIn the introduction, the authors should investigate the effect of bisphosphonates on OI.

In 2.2 and 2.3 the authors should also report other studies on the validation of mSCs in OI.

Paragraphs 3.1 and 3.2 can be simplified with a cartoon

Did the authors investigate the role of anticholesterolemic drugs (such as statins) in OI?

The authors could complete the Review with a graphical abstract.

Comments on the Quality of English Language

Minor editing of English language required

Reviewer 2 Report

Comments and Suggestions for Authors

In this review article, the authors focused on MSC therapy for OI and discussed the potential effects of enhancing the mitochondrial function of MSCs with compounds on OI bone phenotypes. Because many previous studies have demonstrated that enhancing mitochondrial function can improve the osteogenic differentiation of MSCs, compounds that enhance mitochondrial function may benefit OI patients receiving MSC therapy. However, the authors did not consider a critical point. When treating genetic diseases such as OI, MSCs must be administered systemically, not locally, as all bones are affected. Local or intrauterine MSC injections have shown that some injected MSCs can engraft and differentiate into osteoblasts in bone. However, when injected intravenously, MSCs barely engraft into bone. This is not just a case of OI but a common problem in MSC therapy due to the trap in the lung. Although the precise mechanisms underlying the therapeutic effects of systemic infusion of MSCs in OI remain unknown, the currently available data suggest that direct differentiation of infused MSCs into osteoblasts is unlikely. Rather, paracrine effects of MSCs, including secretion of growth factors/cytokines and/or release of extracellular vesicles, seem to play an important role. Therefore, focusing on osteogenic differentiation of MSCs by enhancing mitochondrial function will have less impact on OI patients when MSCs are infused systemically. Even if the osteogenic differentiation of MSCs is enhanced, the therapeutic effects would not be improved that much. Other comments are described below.

1.      There are inconsistencies in language throughout the manuscript. For example, the authors use both “diphosphonates” and “bisphosphonates”, and “aging” and “ageing”.

2.      Throughout the manuscript, the references cited often do not support the statements made. For example, in Ln82-83, reference 23 does not mention direct differentiation of MSC into osteoblasts as a mechanism of therapeutic effects. Reference 85 does not support the statements in Ln283-285, Ln288-290, and Ln297-298. This type of misunderstanding or misinterpretation of references is often seen in this manuscript. The authors need to cite references more carefully.

3.      There is currently no evidence that OI is induced by mitochondrial dysfunction as mentioned in Ln193. Reference 85 suggests that mitochondria are also affected, but it remains unknown whether this is a cause or an effect secondary to the accumulation of misfolded collagen in the ER. Thus, we still don’t know if supporting mitochondrial function can be therapeutic in OI.

4.      Related to the above comment, mitochondrial dysfunction in OI has nothing to do with this review because the compounds that enhance mitochondrial function are used for MSCs and not to support the mitochondria of endogenous OI cells. The way the authors wrote throughout the manuscript, it is not clear whether these compounds are used systemically with MSC or just to treat MSCs prior to cell therapy.

5.      Related to comment #1, the authors use MSC and BMSC incorrectly. For example, they started using BMSCs in Ln235, although reference 69 uses MSC instead of BMSC. In Ln290-293, reference 87 uses PO-MSCs (periosteum-derived MSCs), but the authors use BMSCs.

6.      In Ln368-369, how does aKG show promise for OI? The references do not support this statement.

7.      Is the statement in Ln426-428 correct? The authors must provide citations for this statement.

8.      Ln438-441, do increased oxidative stress and inflammation cause impaired osteogenic differentiation of MSCs and defective bone formation in OI? References 124 and 125 do not support this claim.

9.      Ln472-473, has this method demonstrated encouraging outcomes in improving bone repair in OI? Reference 136 does not support this claim.

Comments on the Quality of English Language

Overall English seems to be fine. However, both British and American spellings are found. They must be consistent. 

Reviewer 3 Report

Comments and Suggestions for Authors

Dear authors,

my main comment on your work is that the title does not reflect the content of the manuscript. You give scarce information on the OI pathogenesis, and then dedicate the rest of the text to the mitochondria's physiology, function, regulation etc. The only connection between the OI and the content of the text are MSC, as they do posses mitochondria and MSC's function depends on the oxygen level. However, this link is too weak and feeble. The title or the content of the article should be changed.

When you do mention some attempts to treat OI with MSC, you give only vague descriptions. Please, add some details, like the source of MSC, the method of implantation etc.

Comments on the Quality of English Language

Please, check thoroughly that you introduce acronyms only once in the text. OXPHOS, for example, has been introduced twice (lines 208 and 232).

The sentence on lines 278-280 seems incomplete.

There are a few punctuation typos as well. 

I do not have any principal comments on English quality.

Round 2

Reviewer 1 Report

Comments and Suggestions for Authors

The authors have significantly improved manuscript "Advancing Osteogenesis Imperfecta Treatment: The Role of Mitochondrial Homeostasis in Mesenchymal Stem Cells Therapy", so it is now considered publishable in pharmaceuticals journal.

Author Response

Thank you for your previous valuable advice, which helped us to better complete the writing of this article!

Reviewer 2 Report

Comments and Suggestions for Authors

Comments to the authors

The authors have revised the manuscript to address the critiques raised by the reviewers. However, they still focus on the osteogenic differentiation of MSC as the mechanism of action of MSC therapy, which does not contribute significantly to the therapeutic effects of MSC therapy for OI. Although the authors revised the content on paracrine effects in Section 2, the rest of the manuscript, such as Sections 3 and 4, only focus on the osteogenic differentiation of MSC and fail to comment on how mitochondria regulate the paracrine effects of MSCs. It is not mitochondrial problems in MSCs that prevent infused MSCs from engrafting into bone as osteoblasts and regenerating bone by providing normal type I collagen. Therefore, even if you improve mitochondrial function in MSCs, they will not engraft and regenerate bone after systemic infusion. As mentioned previously, this is not just a case of OI, but a common problem in MSC therapy. When MSCs are injected locally into bone, enhanced osteogenic differentiation of MSCs could help to regenerate bone. However, since OI is a genetic disorder that affects all bones in the body, it is not realistic to inject MSCs into every single bone. Therefore, the current version of this manuscript, which focuses on how to enhance the osteogenic differentiation of MSCs, does not address the current challenge in MSC therapy for OI and may mislead readers as if lack of osteogenic differentiation of MSCs is the problem in MSC therapy for OI. Thus, this reviewer recommends that the manuscript be extensively revised to either focus on mitochondrial regulation on paracrine effects of MSCs in MSC therapy for OI or forget about OI and focus only on regenerative medicine using MSCs with enhanced osteogenic capacity by targeting mitochondria.   

(Previous Comment #3): It appears that nothing was deleted. It still says that “Mitochondrial dysfunction can lead to range of diseases, including not only OI, but also aging, cancer….”. As mentioned in the previous comment, there is currently no evidence that OI is induced by mitochondrial dysfunction.

(Previous Comment #4): As mentioned, it is not clear whether the compounds are used just to treat MSCs prior to cell therapy or whether they are injected systemically. In the references cited in the Vitamin C and a-LA section, these compounds were used systemically, not to treat MSCs. If compounds are used to treat MSCs, they will not affect the mitochondrial dysfunction in the cells of  OI patients. If they are injected systemically, they could affect the mitochondrial function of OI patients’ cells, but then the title does not reflect that.     

(Previous Comment #6): Again, enhancing osteogenic differentiation of MSCs does not overcome the current challenge of MSC therapy for OI because infused MSCs cannot engraft into bone. Thus, it doesn’t show promise as a therapeutic treatment for OI. The two references (85 and 105) provide no evidence to support this promise.

(Previous Comment #7): There is no evidence or previous report showing that a-LA enhances bone regeneration in OI. None of the references cited by the authors here evaluate the effects of a-LA in OI. So you cannot say “a-LA therapy enhances bone regeneration in OI patients……”. 

Comments on the Quality of English Language

Overall English seems to be fine.

Reviewer 3 Report

Comments and Suggestions for Authors

Dear authors,

thank you for the additions to the text. It seems more adequate to its title now. 

Comments on the Quality of English Language

Please, check the acronyms. MSC are introduced twice (lines 11 and 291) at least.

Author Response

Thank you for your previous valuable advice, which helped us to better complete the writing of this article. The issue with the acronyms has been checked and corrected.

Round 3

Reviewer 2 Report

Comments and Suggestions for Authors

The authors reason that regulation of osteogenic differentiation of MSCs is critical for MSC therapy for OI based on references 48, 49, 50, and 12. These references showed that the osteoblastic engraftment of donor cells was quite low, not exceeding 2%, despite the clinical improvements. In particular, engraftment after BMSC transplantation was less than 1% as shown in reference 50, which was worse than after bone marrow transplantation as demonstrated in references 48 and 49. These results suggest that the clinical improvements are less likely due to the engraftment of BMSCs. Indeed, the Horwitz group also thought that it was nonadherent bone marrow cells such as hematopoietic cells rather than BMSCs that engrafted as osteoblasts after transplantation. Thus, in their follow-up studies, they sought to investigate the possibility that nonadherent bone marrow cells become osteoblasts after transplantation (Dominici et al., PNAS 2004, Dominici et al., Blood 2008 and 2009, Otsuru et al., Blood 2012). In addition, this group has demonstrated that growth acceleration can be achieved using extracellular vesicles of BMSCs without BMSCs themselves. Therefore, the current understanding of the mechanism underlying the therapeutic effects of MSC therapy for OI in the field is not the osteogenic differentiation of infused MSCs. There is no scientific premise that enhancing the osteogenic differentiation of MSCs can improve the osteoblastic engraftment after MSC transplantation. Therefore, this review would mislead readers as if the lack of osteogenic differentiation of MSCs is the current challenge in MSC therapy for OI.
